# Effectiveness of Online Programmes for Family Carers of People with Intellectual Disabilities: Systematic Review of the International Evidence Base

**DOI:** 10.3390/healthcare12131349

**Published:** 2024-07-06

**Authors:** Rachel Leonard, Nathan Hughes, Trisha Forbes, Michael Brown, Lynne Marsh, Maria Truesdale, Stuart Todd, Mark Linden

**Affiliations:** 1School of Nursing and Midwifery, Queen’s University Belfast, Belfast BT7 1NN, UK; rachel.leonard@qub.ac.uk (R.L.); m.j.brown@qub.ac.uk (M.B.); l.marsh@qub.ac.uk (L.M.); m.linden@qub.ac.uk (M.L.); 2Department of Sociological Studies, University of Sheffield, Sheffield S10 2TN, UK; nathan.hughes@sheffield.ac.uk; 3College of Medical Veterinary and Life Sciences, University of Glasgow, Glasgow G12 8QQ, UK; maria.truesdale@glasgow.ac.uk; 4School of Care Sciences, University of South Wales, Usk Way, Newport NP20 2BP, UK; stuart.todd@southwales.ac.uk

**Keywords:** family carers, intellectual disability, systematic, review, online, health and well-being

## Abstract

This systematic literature review examines the evidence base on the effectiveness of online programmes on the mental health and well-being of family carers of people with intellectual disabilities. Databases (ERIC, Medline, PsycINFO and CINAHL) were searched for intervention studies that considered online interventions for family carers of people with intellectual disabilities. Data were extracted using standardised data extraction tools. Bias was assessed using the Cochrane risk of bias tool for randomised trials (RoB 2). Screening, extraction and assessment of bias were completed independently by two members of the research team. Given the low number of included studies and different outcomes assessed within them, it was not possible to conduct a meta-analysis; therefore, data are presented narratively. Two studies met the criteria to be included in the review. Both studies utilised a feasibility randomised controlled trial methodology. One study found a significant decrease in parental stress, while the other found a significant increase in psychological well-being. Caution must be taken in drawing firm conclusions, given the small sample sizes and low retention rates in both studies. Online programmes seem to offer potential benefits to family carers of people with intellectual disabilities. However, further investigation is needed to examine these programmes, adopting a collaborative approach with family carers.

## 1. Introduction

Family carers of people with intellectual disabilities (IDs) provide ongoing care for their family members, which may include medication management, monitoring a physical condition and behaviour management [1]. Family carers play a crucial role in supporting their health and well-being; approximately 77% of people with IDs in England were living with their families in 2017/2018 [2]. The benefits of directly supporting carers were also highlighted by the UK’s Cabinet Office as part of their ‘Think Family’ policy initiative [3,4]. The report recognised the importance of services and supports that address the specific and independent needs of family members providing care to maintain or enhance the support they are then able to provide. This includes supports that only indirectly relate to their caring role, for example, to maintain their paid employment or address their mental well-being. Such services and supports, therefore, perceive the carer as a service user in their own right [5].

Family carers report positive experiences of having a family member with an ID [6]. A systematic review of 22 studies detailed positive increased personal growth, strength and development of parents while acknowledging that distress and stress may still be present for some [6]. There is recognition of the need to promote and support the mental and physical health of family carers of children with IDs. A number of factors have been shown to positively impact the quality of life of families, including informal support from family and friends, access to professional supports, social relationships and, for some, spiritual and religious support [7]. Other factors found to effect family carers’ health and ability to provide care and support include characteristics of the child and family and access to support and advice [8]. Family carers of people with IDs are twice as likely as other carers to experience stress and mental health challenges; however, there is a lack of research exploring suitable interventions [9].

While recognising the positive effects of having a family member with an ID, some family carers experience significant and ongoing stress and mental health difficulties [10,11]. A UK-based study interviewed 18 mothers who cared for children with IDs with challenging behaviours, finding they experienced social isolation, conflict, limited lifestyles and self-blame [12]. Mothers highlighted that they were surviving day to day and described conflict with professionals over a lack of service provision [12]. The mothers were also shown to possess higher levels of anxiety and depression in comparison to a group of mothers with typically developing children [13]. The demands of caregiving and the lack of appropriate support also make participation in employment difficult, particularly for mothers [14]. There is less available research on the impact of caregiving on fathers; however, a qualitative study described how fathers’ identities changed following diagnosis from that of ‘father’ to ‘father with a child with ID’; this unanticipated change would affect their relationship across the child’s lifespan [15]. A systematic review and meta-analysis suggested that fathers have lower mental health difficulties, possibly due to their more limited caring role [16]. A second review identified the emotional consequences of a diagnosis of ID, finding that fathers experienced feelings of grief, loss, denial and blame [17].

Programmes and interventions which provide training to family carers have been shown to have a positive impact on levels of stress and feelings of confidence [18], child socialisation [19] and quality of life [20]. However, parents face barriers in taking part in such interventions, including time, family pressures, cost and availability of services [21]. Online programmes offer greater flexibility of delivery and have been shown to offer comparable outcomes to more traditional treatment approaches in areas such as parenting styles [22], knowledge and self-efficacy [23]. By online programme, we refer to any intervention or training delivered via the internet for the purposes of improving the lives of carers of people with ID. 

The COVID-19 pandemic made access to face-to-face support even more difficult for families, with attempts being made by some services to offer online alternatives [24,25]. While this has created challenges for family carers (e.g., access to reliable technology), it has also provided opportunities (e.g., reduced travel and delivery costs, improved access to programmes). In one study, mothers reported the need to stay healthy and how they made the changing pandemic situation work for their child with ID [26]. Families reported increased levels of stress and burden of care because of the lockdowns, in part due to the closure and changes to service provision, resulting in significant additional family pressures [27]. Family carers of adults with ID also reported receiving limited support from services and experienced feelings of powerlessness. However, support from family and friends was helpful and resulted in resilience [28]. A recent scoping review evidenced the importance of learning from the COVID-19 pandemic from the perspectives of people with ID and their family carers [29]. The review identified three core themes related to ensuring that preventative information is quickly available and accessible, that reasonable adjustments are made in response to need (e.g., access to technology to facilitate the provision of care and support) and that treatment and management needs are identified and enabled (e.g., access to acute hospitals and lifesaving interventions) [22]. 

The purpose of the current work is to review the evidence base on the effectiveness of online programmes on the mental health and well-being of family carers of people with ID. 

## 2. Materials and Methods

The review follows the Preferred Reporting Items for Systematic Reviews and Meta-Analyses (PRISMA) guidelines [30].

### 2.1. Eligibility Criteria 

To be included, studies needed to be randomised controlled trials (RCTs) of online interventions for family carers of people with IDs. Only studies that reported on carers of people with IDs were included. Where a study reported on a sample that included carers of people with IDs and carers of people with other conditions (autism, ADHD etc.), we followed the predominance rule. Namely, to be included, the sample must have predominately been made up of carers of people with IDs (i.e., over 50%). Studies had to report on carer-focused outcomes to be included. Publications reporting on peer-reviewed empirical studies were included. Only publications available in the English language were included due to resource restrictions. 

The review excluded studies that presented interventions that were not delivered online, reported on carers of people with conditions that were not IDs or did not report on specific carer outcomes (e.g., where the outcome was in relation to the person with the ID, not the carers themselves). Feasibility and pilot studies that did not employ an RCT methodology were not included. Reviews, protocols and discussion articles were also excluded. Empirical studies published in languages other than English were excluded.

### 2.2. Information Sources 

Searches were conducted using the electronic databases ERIC, Medline, PsycINFO and CINAHL.

### 2.3. Search Strategy 

Database choices and search terms were refined through an iterative process including consultation with a subject specialist librarian. Searches used a combination of MeSH (medical subject heading) terms and free text terms, which were organised into categories representing randomised controlled trials, online delivery, programmes, family carers and IDs. These categories related to the research question, which sought to examine online interventions for family carers of people with IDs. Each category was broken down into search terms which represented keywords and phrases used in our searches. Searches were conducted on 18 January 2024. The search terms are presented in Table 1, and an example of the search string in Medline (Ovid) is available in the Appendix A.

### 2.4. Study Selection 

The online platform Covidence [31] was utilised for organising and selecting literature for the review based on the criteria outlined above. Titles and abstracts of all papers were independently screened against the eligibility criteria by two reviewers. Following this, full texts were retrieved for all potentially relevant studies, which were then screened independently by two reviewers at the full-text level. All conflicts between reviewers were resolved by discussion, with a third reviewer acting as the arbiter. The process is outlined in the PRISMA diagram in Figure 1 [30].

### 2.5. Data Extraction 

Data were extracted in relation to country, study aim(s), study design, sample, intervention description, comparison, outcomes measured, findings for the primary outcome and conclusions. Data extraction was performed independently by two members of the review team using standardised data extraction tools.

### 2.6. Risk of Bias Assessment 

Two reviewers independently reviewed the risk of bias of the included studies, discussing any discrepancies. The revised Cochrane risk of bias tool for RCTs (RoB 2) [32] was employed to assess bias in the included studies. The RoB 2 contains five domains through which bias is assessed: (1) bias arising from the randomization process; (2) bias due to deviations from intended interventions; (3) bias due to missing outcome data; (4) bias in the measurement of the outcome and (5) bias in selection of the reported result. Each domain contains signalling questions which elicit information relevant to an assessment of risk of bias. The responses include yes; probably yes; probably no; no and no information. These answers feed into algorithms which determine the bias within each domain and overall. The possible risk of bias judgements are: (1) low risk of bias; (2) some concerns and (3) high risk of bias.

### 2.7. Outcome of Interest 

This review primarily focused on measures of carers’ mental health and well-being. Well-being and mental health were measured according to the mean overall change (from baseline to endpoint) assessed by any psychometrically validated tool. Examples of tools include the Warwick–Edinburgh Mental Well-being Scale (WEMWBS) [33] and the Parenting Stress Index—Short Form (PSI) [34].

### 2.8. Data Analysis and Synthesis 

Due to the low number of studies and differences in measurements used, it was not possible to conduct a meta-analysis; therefore, the results are reported narratively [35]. This allowed key concepts to be identified and grouped together for ease of comparison. Concepts were grouped under subject headings or themes, which were then discussed by members of the research team for verification.

## 3. Results

As seen in Figure 1, our searches yielded a total of 718 results. After removal of duplicates (*n* = 111), screening titles and abstracts, we were left with 17 results. Of these 17 records, 15 were excluded. The reasons included the following: the intervention was not conducted online (*n* = 5); the study was not an RCT design (*n* = 4) and study did not predominantly consider IDs (*n* = 6). As such, two studies were included in this review.

### 3.1. Study Characteristics 

Details of the two studies that met our inclusion criteria are displayed in Table 2. One study conducted a randomised feasibility trial [9], with the other conducting a randomised effectiveness trial [36]. Flynn et al.’s [9] study was conducted in the UK, while Grenier-Martin et al. [36] conducted their trial in Canada. Both studies utilised a comparator, one with a waitlist control (intervention received at week 4) [36] and the other compared against the same intervention but with an additional element (x3 30 min peer support telephone calls). Samples ranged in size from 42 [36] to 60 [9]. Follow-up rates of completed outcome measures were 83% (50/60) [9] and 69% (29/42) [36]. Both studies included family carers of children (<18 years) with IDs. In Flynn et al.’s [9] sample, most children had a diagnosis of ID (*n* = 54), with some having a dual diagnosis of ID and autism (*n* = 42). Carers themselves were predominantly female (*n* = 55 out of 60) and white British (*n* = 48 out of 60) [9]. Grenier-Martin et al. [36] did not report on the age or sex of their participants. Many of the children of the carers had a chromosomal abnormality or genetic syndrome (72.4%); the remaining children had an ID (24.1%) or cerebral palsy (3.4%) [36].

### 3.2. Intervention Description 

Flynn et al. [9] tested the feasibility of the ‘Be Mindful+’ online intervention. Be Mindful is a publicly available online mindfulness programme which includes ten online sessions based on the elements of Mindfulness-Based Cognitive Therapy. The sessions are presented by two qualified mindfulness trainers. Participants are provided with twelve assignments to practice in daily life and six downloadable course handouts, and auto-generated supporting motivational emails were also included throughout [9]. The study aimed to test an additional element to the Be Mindful programme; a peer support element consisting of three 30 min telephone calls from a Peer Mentor (a carer who had been through the Be Mindful programme). Participants were given four weeks to complete the online intervention. 

Grenier-Martin et al. [36] tested the effectiveness of an online training programme to support carers in managing problem behaviours at home. The online training was asynchronous (i.e., parents were able to learn at their own pace) and consisted of five separate modules. At the end of each module, quizzes evaluated parental comprehension. Modules included: how to document and assess the functions of problem behaviours, what an antecedent was, the different intervention strategies associated with antecedents, the influence of the consequences on the behaviour, how caregivers should react to problem behaviours, methods for teaching appropriate behaviour as an alternative to problem behaviour and practical considerations for the intervention. Participants were given two weeks to complete the online intervention.

### 3.3. Primary and Secondary Outcomes 

While both studies included carer outcomes, only Flynn et al. [9] examined these as the primary outcome. The primary outcome of Grenier-Martin et al. [36] was to evaluate the effects of this online training on the intensity and frequency of problem behaviours observed in children by their caregivers, as measured by the Adaptive Behaviour Scale Assessment System—2nd Edition (problem behaviours in children) [37] and the Behaviour Problems Inventory-01 [38]. Secondary outcomes included the Parenting Stress Index—Short Form [34] and the Self-Assessment Questionnaire of the Parent’s Educational Competence [39]. Flynn et al. [9] primarily measured psychological well-being, as measured by the Warwick–Edinburgh Mental Well-being Scale (WEMWBS) [33]. Secondary outcomes included quality of life, psychological distress and family functioning. 

Grenier-Martin et al. [36] completed measures over the phone or at home with two psychology PhD students, depending on location and the caregiver’s preference. Outcomes were assessed at four time points: Week 1, 6, 10 and 14 for the intervention group and week 1, 4, 10 and 14 for the control group [36]. Flynn et al. [9] assessed outcomes at three time points: baseline, 12 weeks and 6 months post-randomisation. 

Grenier-Martin et al. [36] only provided data for the Parenting Stress Index—Short Form (PSI) [34] at T2 for the intervention and control groups. In their statistical analysis, they controlled for T1 results; however, they did not provide comparable data for T1, T3 or T4. It is also important to note that the time points for T2 were different for the intervention and control groups. Flynn et al. [9] compared scores on the WEMWBS (psychological well-being) 6 months post-randomization, adjusting for baseline scores. Data were not presented for outcomes at week 12.

### 3.4. Risk of Bias 

#### 3.4.1. Bias Arising from the Randomisation Process

Both studies utilised random allocation, with concealment of allocation until after participants were assigned to the intervention or control conditions. Grenier-Martin et al. [36] used block randomisation, with a spreadsheet automatically assigning participants to the experimental or waitlist groups. Flynn et al. [9] stated that participants were randomly allocated, but did not provide details on randomisation procedures. No significant differences were noted in baseline demographic information for either study. Both studies were rated as having a low risk of bias within this domain.

#### 3.4.2. Bias Due to Deviations from Intended Interventions

With both studies, it was not possible to blind participants to allocation. However, the researchers were blinded throughout data collection in both studies [9,36]. Adherence to intervention was poor within Flynn et al. [9], with 13 out of 30 fully completing ‘Be Mindful’ and 14 out of 27 fully completing Be ‘Mindful+’ (52%). In addition, only 3 of the intervention group completed ‘Be Mindful+’ within the timeframe, with 23 out of 30 receiving peer support telephone calls. Grenier-Martin et al. [36] did not specifically discuss intervention adherence; however, they had a 69% (29/42) follow-up rate. Specifically, this was 16 out of 22 for the control group and 13 out of 20 for the intervention group. Neither Flynn et al. [9] nor Grenier-Martin et al. [36] accounted for this loss to follow up or lack of adherence to intervention within the analysis of data. It is important to note that Flynn et al. [9] were primarily concerned with feasibility outcomes as opposed to the establishing effectiveness of the intervention on carer well-being. Within this bias domain, Flynn et al. [9] and Grenier-Martin et al. [36] were rated as having ‘some concerns’. 

#### 3.4.3. Bias Due to Missing Outcome Data

Loss to follow-up was high in both studies, at 83% (50/60) [9] and 69% (29/42) [36]. Missing outcome data was largely due to participants withdrawing from the study. Reasons for withdrawal were provided by Flynn et al. [9]; however, no details were provided by Grenier-Martin et al. [36]. Neither study used analysis methods that mitigated the risk of bias from missing data (i.e., Intention to Treat (ITT) or analysis based on ITT principals). For example, participants were not excluded based on a minimum amount of the completed intervention. Flynn et al. [9] analysed 50 out of 60 participants who had provided outcome data. However, only 27 out of 60 fully completed the intervention (both control and intervention groups). Grenier-Martin et al. [36] did not provide details on intervention adherence, so it is not possible to determine whether they excluded participants from their analysis based on the minimum number of participants who completed the intervention. For both studies, the proportions of missing outcome data were similar across groups. Both studies were rated as having a low risk of bias within this domain. 

#### 3.4.4. Bias in Measurement of the Outcome

Both studies utilised self-reported questionnaires (PSI and WEMWBS). Outcome assessors remained blinded throughout data collection. In addition, the comparator in both studies was another active intervention (waitlist control and Be Mindful), which was facilitated by an independent researcher outside of their standard care. Both studies were rated as having a low risk of bias within this domain. 

#### 3.4.5. Bias in Selection of the Reported Results

Both studies either failed to report or partially reported outcomes that were measured. Flynn et al. [9] did not report on week 12 data, reporting only a subset of time points (baseline and 6 months post-randomisation). On this basis, Flynn et al. [9] was rated as having ‘some concerns’ for this domain. Grenier-Martin et al. [36] failed to report T1, T3 or T4 individual group data. The only data reported were a comparison of T2 scores; however, these were collected at different time points for the intervention and control groups (week 4 and week 6). For the other time points, Grenier-Martin et al. [36] reported on combined scores for control and intervention groups. It is unclear why the authors opted for this analysis, as it limits the conclusions that can be drawn regarding the effect of the invention. On this basis, Grenier-Martin et al. [36] was rated as ‘high risk of bias’ for this domain. 

#### 3.4.6. Overall Judgement of Bias 

Due to having concerns around deviations from intended interventions and selection of reported results, Flynn et al. [9] was assessed as having ‘some concerns’ in relation to the overall risk of bias judgment. Grenier-Martin et al. [36] had concerns around deviations from intended interventions and a high risk of bias within the selection of the reported results domain. Overall, Grenier-Martin et al. [36] was assessed as having a ‘high risk’ of bias. Each domain and judgment is presented in Figure 2 and Figure 3.

### 3.5. Effectiveness and Impact of Online Interventions for Carers 

The two studies offer some indicative evidence of the potential impact of online interventions for carers of people with ID, though caution must be taken in drawing firm conclusions. Grenier-Martin et al. [36] demonstrate that their online training for managing problem behaviours produced a significant decrease in parental stress from pre-training to follow-up (F (2,56) = 5.26, *p* < 0.01). This significant decrease was immediately apparent post-intervention (*p* = 0.024) and was sustained over time, with follow-up measures taken up to 2 months. However, the sample size was small, and the attrition rate was 30%. 

Flynn et al. [9] indicated significant increases in psychological well-being (β = 1.15, *p* < 0.001) following the Be Mindful intervention. However, all participants in the study received the online intervention, and so the study design did not test the intervention against no online support. Instead, the study tested the value of an additional telephone support service, providing ‘space for reflection’ for carers undertaking the online programme. The study ‘was not powered to detect differences in outcomes between the two arms of the trial’, and found ‘small, but not significant, improvements’ for those who received the additional telephone support.

The two studies also offer some indication that online interventions may be suitable and attractive to carers of people with IDs. Flynn et al. [9] highlighted the difficulties for carers of people with IDs to prioritise their own mental well-being given competing commitments associated with their care. Online provision is seen to offer ‘a potential solution’, given that it can be completed alongside other commitments in a flexible way. Similarly, Grenier-Martin et al. [36] reported that participants valued the approach of online delivery, and that the content of such a programme was consistent with their needs. The authors also reflected on the potential for online support to provide immediate and timely support, accessible to all, at a time when families may wait months or even years for specialized intervention services. 

### 3.6. Challenges in Delivering Interventions for Carers 

The two studies suggest some challenges in the use of online support for carers of people with IDs. Most notably, both studies struggled with retention. Grenier-Martin et al. [36] suffered from a 30% attrition rate, and such rates are common among online programmes for parents. 

Both studies suggest potential limitations in reliance on online content and value in additional human contact, whether with professionals or peer mentors. Flynn et al. [9] specifically examined the value of additional telephone support from a peer mentor, providing space for reflection about their completion of the course. They argue that this enables an intervention to become tailored to the individual carer or family in a timely and cost-effective manner. While Grenier-Martin et al. [36] did not include such a mechanism, they also argued for further research into the relevance of enabling carers to engage remotely with professionals during the online training.

## 4. Discussion

This review sought to synthesise robust experimental research regarding the effectiveness of online interventions on the well-being and mental health of family carers of people with ID. However, only two studies met the eligibility criteria, and both presented limitations in relation to dimensions of study quality and potential bias, including high attrition rates. This indicates a dearth of high-quality evidence in this field. 

Both studies highlight some potential benefits of using online approaches, including the flexibility with which online material can be engaged with, so as to meet other commitments, and the efficacy of online delivery of the specific information needs of family carers. These are in keeping with the benefits noted in studies of online interventions for carers of people with other forms of support needs [40,41,42]. For example, other studies have also noted the benefits of being able to access online content when and where it most suits a carer’s other commitments [40,41], while the benefits to carers of video recordings that can be watched repeatedly and of video conferencing methods have also been [41,42]. Other benefits reported elsewhere include addressing inaccessibility due to geographical barriers, including for those in remote, rural or isolated settings [41,43,44,45], and reducing the participation costs of childcare or transportation for carers [41,46,47] or barriers to participation due to lack of access to transport [45].

Both studies also highlight some of the potential limitations to, or challenges with, online provision for carers of people with IDs. Recruitment and retention rates may be indicative of challenges in engaging carers sufficiently to complete online programmes. Both studies also suggest that there is value in additional personal contact alongside online content delivery, and it has been argued elsewhere that online provision might be most useful as a complimentary approach to traditional approaches to supporting carers, rather than simply a replacement [42,44]. While not prominent in the two studies reviewed here, other studies have suggested potential challenges in carers’ access to and difficulties with technology. Online programmes are clearly limited to those families and carers who own the necessary technology and sufficient internet connection [45,48,49], and the latter may be a particular issue for carers in rural areas [47]. Technical issues are also reported elsewhere, including web browser issues and problems with logging in [46,47], which have been seen to disrupt the fluidity of collecting and observing data [48]. However, as we become increasingly used to mobile devices and have better access to reliable digital networks, many of these difficulties will be overcome.

There is growing recognition of the utility of online approaches for addressing the needs of family carers, which may partly be due to the increasing digital literacy of carers and the ubiquity of technology. The COVID-19 pandemic has significantly increased familiarity with online platforms [25] and demonstrated that many services can be delivered online. Of course, some services will always need to be delivered face to face, and online platforms should be used where they can be most effective in addressing the needs of carers. Overall, our findings suggest merit in further research to establish an evidence base for the use of online support for carers of people with ID. Funding bodies should prioritise the development and testing of high-quality interventions to support family carers in their crucial role in providing care to their family members.

### 4.1. Implications for Policy

Arising from the findings of this review are implications for policy that need to be considered for the future, notably in relation to long COVID and the longer-term effects of lockdown. As the full implications and long-term effects of the COVID-19 pandemic are realised, policy makers need to ensure that the needs of families caring for people with ID are fully recognised and included. This is important due to the impact on the health and well-being and quality of life of families when caring for a family member with an ID [50,51]. There is scope for policy makers to ensure that the specific needs of families of people with ID are included in post-pandemic responses, thereby seeking to ensure that there is ongoing access to social support networks [52]. As the full extent of the effects of COVID-19 and lockdown restrictions during the pandemic on physical and mental health of family carers becomes apparent, primary care and mental health policies need to reflect the needs and concerns of families and their family members with IDs [53,54]. Online programmes for family carers may be one approach for inclusion in future health policies. Failure to recognise and respond to the needs of this population will negatively impact their long-term health and their capacity and capability to continue to care and provide support for their family members [55].

### 4.2. Potential Biases in the Review Process

We reduced bias in this review by following predetermined data extraction and risk of bias questions, and ensured our search terms were appropriate through consultation with a subject specialist librarian. We did not seek to include studies published in languages other than English. This was due to limitations in the availability of translation services and means that we may have missed some aspects of the international evidence-base.

### 4.3. Suggestions for Future Research

This review found only two RCTs amongst the available literature. This suggests a need for large-scale, evidence-based interventions that have been thoroughly tested prior to implementation. No studies in this review were conducted in low- or middle-income countries (LMIC), and there was limited participation from ethnically diverse carers. Online delivery of interventions to support family carers seems ideally placed for use in LMIC due to their relatively low cost and long reach. However, the reliability, or existence, of internet connections in certain regions may mean that efforts to introduce such programmes would be stymied. With improvements in technologies such as mobile internet or satellite-based services, such barriers could be overcome. Future research should seek to understand the barriers and facilitators to delivery in LMIC and create interventions which can address these whilst supporting family carers.

A related area for future investigation would be the exploration of rural versus urban differences in the delivery of interventions. Family carers living in rural locations in many developed regions of the world experience reduced services and a lack of access to care. Online programmes have been successfully used in remote locations across the world; however, few of these have been rigorously evaluated. Future research should seek to conduct RCTs of these programmes to determine whether they are effective.

In the post-COVID era, many services have returned to in-person delivery. Many family carers prefer this approach; however, for some, the demands of their caring role mean that regular attendance can be difficult to manage. In such cases, online delivery of services could be seen as a means to enhance, or compliment, existing services rather than a replacement. This blended approach to service delivery could be examined to identify successful components to better refine delivery and provide the optimal approach which best suits the needs of family carers.

## 5. Conclusions

More research focusing on online support programmes for family carers of those with IDs is needed. This work should be co-designed with family carers so that it addresses issues which directly impact their lives. Much of the work included in this review was conducted in high-income countries; however, the ubiquity of online platforms and mobile technology would mean that people living in low-income countries could also benefit from inclusion in such work. It may not be appropriate to deliver all types of support and services online; however, such programmes offer a low-cost opportunity to deliver care and support which is convenient and open to all who wish to take advantage of it.

## Figures and Tables

**Figure 1 healthcare-12-01349-f001:**
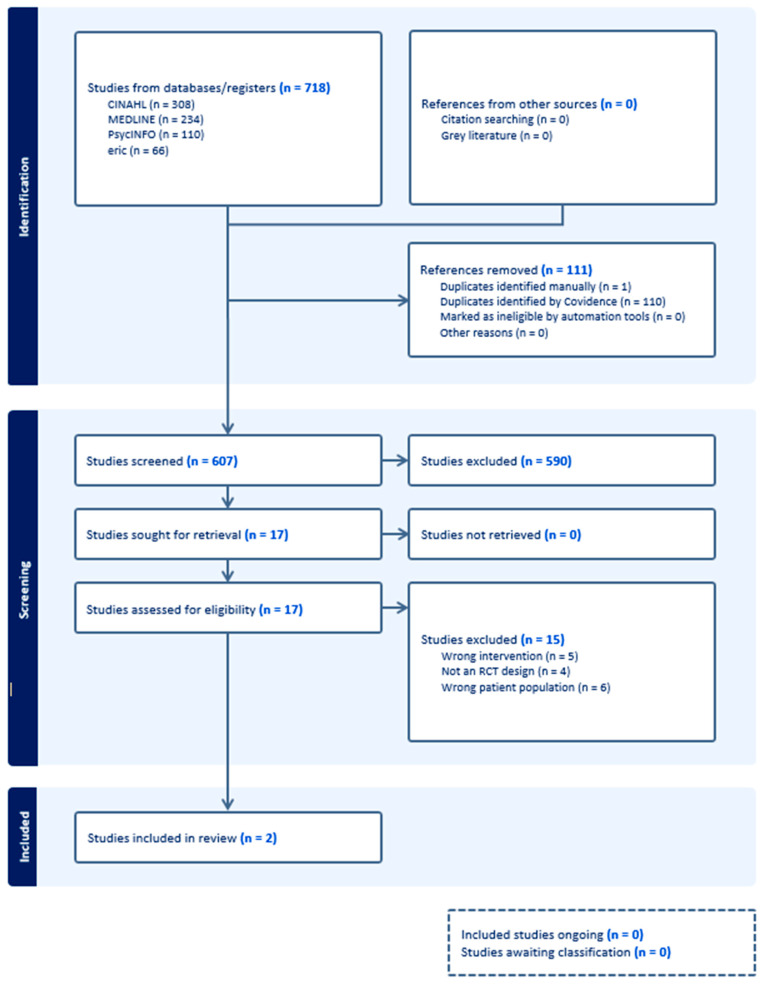
PRISMA diagram.

**Figure 2 healthcare-12-01349-f002:**
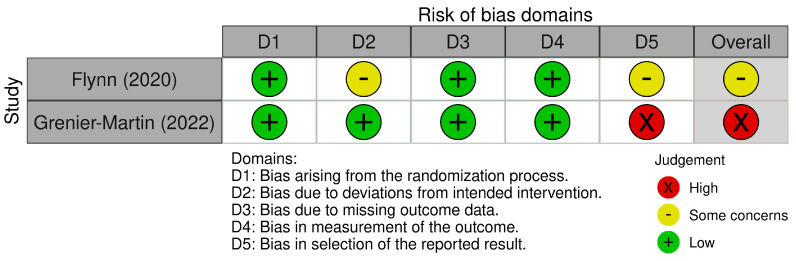
ROB2 domains [9,36].

**Figure 3 healthcare-12-01349-f003:**
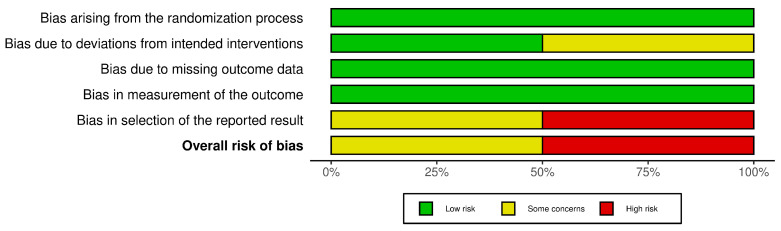
ROB2 overall judgment.

**Table 1 healthcare-12-01349-t001:** Search terms.

RCT	Family Carer	Online	Programme	Disability
(“adaptive clinical trial” or “clinical trial” or “clinical trial, phase i” or “clinical trial, phase ii” or “clinical trial, phase iii” or “clinical trial, phase iv” or “controlled clinical trial” or “equivalence trial” or “multicenter study” or “pragmatic clinical trial” or “randomized controlled trial”).pt. or double-blind method/or “adaptive clinical trials as topic”/or “clinical trials as topic”/or “clinical trials, phase i as topic”/or “clinical trials, phase ii as topic”/or “clinical trials, phase iii as topic”/or “clinical trials, phase iv as topic”/or “controlled clinical trials as topic”/or “equivalence trials as topic”/or “intention to treat analysis”/or “non-randomized controlled trials as topic”/or “pragmatic clinical trials as topic”/or “randomized controlled trials as topic”/or “multicenter studies as topic”/or evaluation studies/or “evaluation studies as topic” or program evaluation/or validation studies/or “validation studies as topic” or (effectiveness or intervention or (pre-adj5 post-) or (pretest adj5 posttest) or (program* adj6 (evaluate or evaluated or evaluates or evaluating or evaluation or evaluations or evaluator or evaluators)	(Care* or Parent* or Famil*)	(Online or Internet or E-learning or Web* or Virtual or Computer-based or Electronic or Tech* or Telemedicine)	(Program* or Intervention or Training or Education*)	(Intellectual disab* or Learning disab)

**Table 2 healthcare-12-01349-t002:** Characteristics of included studies.

Author and Country of Origin	Aim(s)	Design	Participants	Intervention Description	Intervention Duration	Comparison	Primary Outcomes and Time Points	Findings
Flynn (2020) UK	To examine whether Be Mindful can be delivered successfully to family carers of children or adults with ID, and whether it would be feasible to conduct a later definitive randomized controlled trial of the effectiveness and cost-effectiveness of Be Mindful.	Feasibility randomised control trial—parallel design.	Family carers (*n* = 60) (50 analysed) Female (*n* = 55) Mean age 46.09 (SD7.71) The people for whom the participants cared: mean age of 13.73 (SD 8.97) Diagnoses of ID (*n* = 54) and/or autism (*n* = 41).	Be Mindful is a publicly available online mindfulness programme developed by the Mental Health Foundation, which has ten easy-to-follow online sessions based on the elements of Mindfulness-Based Cognitive Therapy. Be Mindful+ participants were offered a further two telephone mentoring sessions—two 30 min telephone calls.	4 weeks plus three 30 min telephone calls	Be Mindful—without telephone mentoring sessions	Psychological well-being was assessed using the Warwick–Edinburgh Mental Well-being Scale (WEMWBS). Baseline, 12 weeks and 6 months post-randomisation	At 6 months post-randomisation, there was a greater increase in psychological well-being for Be Mindful+ compared with Be Mindful, but this was not statistically significant (β = 2.88, *p* = 0.32).
Grenier-Martin (2022) Canada	To evaluate the effectiveness of online parent training for managing problem behaviours for families awaiting specialized service.	Randomised control trial—wait list control design.	Family carers (*n* = 42) (29 analysed) Age not reported Sex not reported The people for whom the participants cared: 4.1 (SD 1.7) Diagnosis of chromosomal abnormality or genetic syndrome (72.4%) ID (24.1%) cerebral palsy (3.4%).	The online training was asynchronous (i.e., parents were able to learn at their own pace) and consisted of five separate modules. At the end of each module, quizzes evaluated parental comprehension.	2 weeks to complete the intervention	Waiting list—control group received intervention in week 4	Parenting Stress Index–Short Form Self-assessment Week 1, 6, 10, 14	Parental stress was reduced for the experimental group as compared to the control group at T2, F(1,26) = 7.63), *p* < 0.05. Across both groups, training produced a significant decrease in parental stress from pre-training to follow-up: F(2,56) = 5.26, *p* < 0.01).

## Data Availability

The original data presented in the study are openly available in Queen’s University Belfast PURE repository at 10.17034/ec9b2b37-8248-40b0-a08a-0f06c4a9bb14.

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
