# Peer review of "Effectiveness of Online Programmes for Family Carers of People with Intellectual Disabilities: Systematic Review of the International Evidence Base"

_healthcare, 2024, doi:10.3390/healthcare12131349_

Round 1

Reviewer 1 Report

Comments and Suggestions for Authors

This is a methodologically sound and well written paper, addressing an important topic.

I suggest amending the title to include ‘Effectiveness of online programmes for mental health and wellbeing of family carers…’.

Typo in abstract ‘Online programmes seem to offer potential benefits. Sound methods clearly described.

Neither study was sufficiently powered for efficacy/effectiveness so important not to over-interpret results; I would like to see more emphasis on the acceptability and feasibility of these approaches., including choice of outcome measure/s in future definitive trials – does either study give much detail about the logic/programme theory underlying the intervention?

Discussion – given the focus of this paper is on effectiveness, it is important to note that both included studies were not sufficiently powered to assess this – and meta analysis was not possible.

Important point made in conclusions re: importance of co-design.

Reviewer 2 Report

Comments and Suggestions for Authors

This is a very nice example of how to conduct this sort of a project, and I am inclined to support publication largely for its instructional value.  Graduate students would be well advised to follow similar processes in literature reviews. 

The limitations of the study are well highlighted in the abstract.  The review process reduced the number of studies that met inclusion criteria to just 2.   I wonder if a less rigorous set of inclusion criteria would have yielded a greater quantity of studies (while obviously reducing quality).   Alternatively, I would be particularly interested in a replication of this analysis using Direct Support Professionals who work in community based group homes.   In addition to the interpersonal measures, I would be interested in measures of DSP retention comparing those who did and did not participate in the training program.

I notice that one of the online techniques involved the use of assignments for participants.   Was there any evidence that the assignments were completed?  

Reviewer 3 Report

Comments and Suggestions for Authors

Thank you for the opportunity to review this systematic review on the effectiveness of online programmes for family carers of people with intellectual disability. This is a well designed, conducted and presented systematic review. Albeit there was a small number of studies identified and I do wonder if a scoping review may have provided a opportunity to identify factors influencing effectiveness? I have made some fairly minor comments/suggestions directly on the document attached for your consideration.
